Efficient geospatial mapping of buildings, woodlands, water and roads from aerial imagery using deep learning

http://orcid.org/0009-0001-0117-4390 Abbas Sidra 1 sidraabbas@ieee.org
http://orcid.org/0000-0002-8665-1669 Almadhor Ahmad 2
Sampedro Gabriel Avelino 3 4
http://orcid.org/0000-0002-6584-7400 Alsubai Shtwai 5
Al Hejaili Abdullah 6
Strážovská Ľubomíra 7 lubomira.strazovska@fm.uniba.sk
Zaidi Monji Mohamed 8
1 Department of Computer Science, COMSATS Institute of Information Technology , Islamabad , Pakistan
2 Department of Computer Engineering and Networks, College of Computer and Information Sciences, Al Jouf University , Sakaka , Saudi Arabia
3 Faculty of Information and Communication Studies, University of the Philippines Open University , Los Baños , Philippines
4 Center for Computational Imaging and Visual Innovations, De La Salle University , Manila , Philippines
5 College of Computer Engineering and Sciences, Prince Sattam bin Abdulaziz University , AlKharj , Saudi Arabia
6 Faculty of Computers & Information Technology, Computer Science Department, University of Tabuk , Tabuk , Saudi Arabia
7 Faculty of Management, Comenius University in Bratislava , Bratislava , Slovak Republic
8 Department of Electrical Engineering, College of Engineering, King Khalid University , Abha , Saudi Arabia
Kong Xiangjie
Electronic publication date: 2024 Jun 25
Publication date: 2024
Volume: 10
Electronic Location ID: e2039
Received 2023 Sep 21; Accepted 2024 Apr 12
Copyright: © 2024 Abbas et al.
Copyright year: 2024
Copyright holder: Abbas et al.
License: This is an open access article distributed under the terms of the Creative Commons Attribution License, which permits unrestricted use, distribution, reproduction and adaptation in any medium and for any purpose provided that it is properly attributed. For attribution, the original author(s), title, publication source (PeerJ Computer Science) and either DOI or URL of the article must be cited.
License URL: https://creativecommons.org/licenses/by/4.0/

Keywords: Land cover, Ariel imagery, Geospatial information, Data augmentation, Deep learning

Funding: Deanship of Scientific Research at King Khalid University RGP2/384/45 This work was supported by the Deanship of Scientific Research at King Khalid University under grant number RGP2/384/45. The funders had no role in study design, data collection and analysis, decision to publish, or preparation of the manuscript.

==============================
As more aerial imagery becomes readily available, massive volumes of data are being gathered constantly. Several groups can benefit from the data provided by this geographical imagery. However, it is time-consuming to manually analyze each image to gain information on land cover. This research suggests using deep learning methods for precise and rapid pixel-by-pixel classification of aerial imagery for land cover analysis, which would be a significant step forward in resolving this issue. The suggested method has several steps, such as the augmentation and transformation of data, the selection of deep learning models, and the final prediction. The study uses the three most popular deep learning models (Vanilla-UNet, ResNet50 UNet, and DeepLabV3 ResNet50) for the experiments. According to the experimental results, the ResNet50 UNet model achieved an accuracy of 94.37%, the DeepLabV3 ResNet50 model achieved an accuracy of 94.77%, and the Vanilla-UNet model achieved an accuracy of 91.31%. The accuracy, precision, recall, and F1-score of DeepLabV3 and ResNet50 are higher than those of the other two models. The proposed approach is also compared to the existing UNet approach, and the proposed approaches have produced greater probability prediction scores than the conventional UNet model for all classes. Our approach outperforms model DeepLabV3 ResNet50 on aerial image datasets based on the performance.

Introduction

Monitoring and assessing land cover and land use are key components of natural resource management. In many sectors, such as urban planning, vegetation monitoring, military reconnaissance, and biodiversity monitoring, comprehensive information about the land cover or land use is a significant resource (Zhou, Huang & Cadenasso, 2011; Pauleit & Duhme, 2000; Ahmed et al., 2017; Gerard et al., 2010). Urban and rural areas have relied heavily on land descriptions provided by satellite data and image processing tools to spot changes. Environment change, woodland dynamics, and destruction can all be inferred from shifts in the landscape (Weber & Hall, 2001; Wickham, O’Neill & Jones, 2000; Potapov et al., 2015; Kennedy, Yang & Cohen, 2010). The rate of urbanization, agricultural intensity, and other human-made changes can be investigated by analyzing landscape dynamics such as fluxes between different land covers (Feranec et al., 2010).

Most studies in this area rely on multispectral aerial imagery, which can be helpful for various applications. Free satellite data normally has a resolution of 10–30 m, but commercial aerial images with a higher resolution can be expensive (Wulder et al., 2012). On the other hand, aerial images are often obtained by local and state governments and can offer pixel sizes as small as 25–50 cm or even smaller. Using aerial imagery to assess land parcel content properly is highly economically important for agricultural and public administration. This is because the government allocates resources and collects taxes based on the intended purpose of the land, be it for agriculture or construction. However, frequent changes in parcel appearance and usage due to human activity require the acquisition of new aerial orthophotos to identify any changes that have occurred. In this regard, classifying and vectorizing all emerging or disappearing objects is necessary. This includes the need for government entities that dole out subsidies to update lists of items that no longer qualify for those incentives. In economics, the most common things that change over time are structures like buildings and trees like forests, rivers, and highways. The usual practice for classification in land cover involves manual methods or simple image classification techniques, such as an object or pixel-based classification (Blaschke, 2010; Khatami, Mountrakis & Stehman, 2016; Caccetta et al., 2016; Ahmed et al., 2017).

Operators examine each orthophoto image in the manual method and mark the “new” objects by hand. This process can take months for the entire country and is highly prone to errors. Our experience indicates an average error rate of over 30% for multiple years over more than 312,000 square km. As a result, manually mapping large land areas is costly (Gerard et al., 2010). As a result, there is a pressing need to create an effective automatic tool to speed up processing while maintaining high precision. A variety of GIS-based software programs offer such tools. The conventional approach in computer vision, which involves manual feature extraction and rule-based methods, must be revised when dealing with high variance and large-scale data. This leads to significant effort and poor scalability. However, because of their scalability, affordability, and high performance, deep learning approaches employing convolutional neural networks (CNN) have become more relevant for automatic classification of aerial images (Khan et al., 2018; Maggiori et al., 2016; Vargas-Munoz et al., 2017). By analyzing the semantic division of certain areas at different periods, CNN enables the rapid and accurate monitoring of considerably wider regions and the classification over time. Unfortunately, big datasets with ground truth annotations are generally necessary for the deep learning strategy. Although aerial images are readily available, the use of computer vision for land cover segmentation is constrained by the difficulty and expense of collecting high-quality annotated datasets. Recently released datasets include some with detailed annotations, while others rely on less precise labeling in training data to identify things like buildings and roads. However, these data sets require methodically segmenting urban areas, forests, bodies of water, and roadways. We present LandCover.ai, a dataset optimized for semantic segmentation, which addresses this shortcoming by including those above four manually annotated classes. In the middle-sized European nation of Poland, we captured images of 216.27 square km of land at 50 cm per pixel and 176.76 square km at 25 cm per pixel. We also supply a model’s baseline findings for use in making judgments. This project aims to develop an automated, highly accurate method of aerial imagery analysis to provide data insights on land cover. This research uses semantic segmentation in computer vision to show how to use deep learning to obtain such insights. This article makes the following contributions. This research contributes to developing an automated system for analyzing aerial images to extract useful information about the land cover.

Augments the data by generating new image samples during training to improve the effectiveness of the proposed model.

Outperformed state-of-the-art research studies and other DL models in a comprehensive evaluation.

The following is the planned framework for studying the categorization of the LandCover.ai dataset: We review the literature on land cover uses classification in “Literature Review” and give an overview of the dataset in “Dataset and Preliminaries”. In “Proposed Approach”, this research can observe that DL is also considered for categorization purposes in this work. In “Experimental Analysis and Results”, we present the findings of this study. “Conclusion” discusses what these results mean and what directions future studies could go.

Literature review

In remote sensing and geospatial analysis, land cover mapping is essential because of the wealth of information it yields for uses, including urban planning, natural resource management, and environmental monitoring. Manual interpretation of aerial imagery is commonly used in conventional land cover mapping approaches, which are both labor-intensive and expensive. Deep learning techniques have greatly aided automatic land cover mapping from aerial imagery in recent years.

For land cover mapping, some studies have turned to deep learning methods. A deep CNN was developed, for instance, by Zhang et al. (2020), to categorize high-resolution remote sensing images. They classified images down to the pixel level using a fully convolutional network (FCN) architecture. They proposed it as a highly efficient, computationally effective method. With the suggested method, high-level features such as spectral patterns, geometric traits, and contextual information are fused to perform per-object categorization. The CRF is then used to fine-tune the per-object categorization result using contextual advice at the pixel level. Five standard approaches, including the current gold standard network DeepLabV3 +, are used to evaluate the approach. Image processing experiments on various datasets show that the previous technique performs exceptionally well for VHRI classification, especially on small datasets. It also demonstrates excellent computing efficiency during the model training and inference processes.

Deep CNNs offer a significant advantage over previous manual methods in speed; however, they require appropriately annotated data. Aerial or satellite photography requires a different perspective and a fine collection of classes than typically used in segmentation datasets developed for common items or street views (Cordts et al., 2016; Lin et al., 2014). Aerial and aerial images are present in some datasets, although these images are often formatted for image classification rather than segmentation. One such example is the UC-Merced dataset (Yang & Newsam, 2010), which was created to classify whole images and contains 21 categories, such as buildings, forests, and rivers, but its resolution of 30 cm per pixel needs to be improved for segmentation tasks. Similarly, Google Maps is the primary source for the image classification datasets (WHU-RS-RSSCN7-AID-NWPU_RESISC45-PatternNet) (Laban et al., 2018; Boguszewski et al., 2021; Xia et al., 2017; Zhou et al., 2018). In order to achieve their potential, deep convolutional neural networks require well-annotated data, as was discussed above. Aerial or satellite photography datasets require a different perspective and an adequate number of classes than those used in traditional segmentation datasets, which often focus on common items or street views. Most aerial and satellite imagery datasets were developed for picture classification. For instance, UC-Merced, one of the earliest satellite datasets, has to be updated for the segmentation task because it is meant for whole-image classification and has a resolution of 30 cm per pixel and 21 categories. There are additional Google Maps picture classification datasets that are very comparable.

However, datasets like DOTA and iSAID are made for multi-class detection and instance segmentation, respectively (Azimi et al., 2019; Waqas Zamir et al., 2019). These datasets have many categories, such as vehicles, bridges, and ships, but they need to be improved for natural resources management, which is a responsibility of public agencies. However, while helpful, datasets developed specifically for building or vegetation segmentation must be more comprehensive for land cover change detection. In addition to Agriculture-Vision, a large aerial image database for agricultural pattern analysis, there are many other databases available, such as the 2010 Tree Cover dataset for the Metropolitan Region of Sao Paulo, the Massachusetts Buildings Dataset, the Inria Aerial Image Labelling Dataset, the AIRS Automatic Mapping of Buildings Dataset, and many more.

Some datasets are based on unmanned aerial vehicles (UAVs), such as ERA, UCLA Aerial Eve, and Okutama-Action; they are designed for event recognition rather than land cover change detection (Mou et al., 2020; Barekatain et al., 2017). The ISPRS Vaihingen and Potsdam datasets are mentioned as manually annotated but are small and have limited coverage of urban areas. Additionally, they need the water class. On the other hand, the Chesapeake Bay Land Cover Dataset (Ortiz et al., 2020) covers a larger area but has a lower resolution and is automatically annotated. While it has many useful classes, it does not include the “building” category. In summary, although several datasets are available for aerial imagery, there is a need for more comprehensive datasets with a broader range of classes, higher resolution, and accurate manual annotation. High-resolution remote sensing footage of both urban and rural areas has been collected, and a new technique is proposed in this article to extract roads from this data (Gao et al., 2019). The technique has two main components: RDRCNN and a TV algorithm for post-processing. Dilated convolution operators form the basis of the RDRCNN model, which is based on ResNet and U-Net architectures. The RDRCNN outputs are combined with the roadways’ line salient features to rejoin the separate areas. The main contributions of this research are the improved CNN architecture for road region detection and the blind voting method for merging regions with misclassification due to visual occlusion. Two difficult datasets are used to test the approach, one created by our researchers. We also compare this strategy to others. While the experimental results show that the method is effective at road extraction from complicated backgrounds (both urban and rural), further processing is needed to draw boundaries more precisely, especially in urban regions.

Wang et al. (2023) proposed DeepDSFusion, a unique decision-level fusion approach based on the Dempster-Shafer (DS) theory. This approach is generally compatible with any other pixel-level segmentation model. Several conventional data augmentation techniques were used in the DeepDSFusion architecture detail to get multiscale probability maps, including rotation and scale transformations. Then, using the DS theory, multiscale probability maps were fused into a single probability map. In order to get segmentation results, a straightforward threshold is finally used in the single probability map. The efficacy of DeepDSFusion is demonstrated by three traditional pixel-level segmentation tasks using high-resolution imagery: landcover mapping, road extraction, and deforestation identification. In Amritesh et al. (2023), the authors thoroughly analyze the outcomes of UNet and QGIS for land-use categorization at higher zoom levels using CNN approaches like UNet. They provide a UNet approach that works remarkably well in Indian circumstances. The solution is excellent for zoom level for cities and neighborhoods inside a city, which is very helpful for appropriate planning. In order to recognize roads and gather information on the different kinds of buildings in the region, they have also connected the UNet model with QGIS. These integrations will help their model identify and anticipate prospective growth areas. The author proposes dividing the landscape retrieval work into smaller projects to pinpoint distinct ideas present in satellite photos (Karatsiolis, Padubidri & Kamilaris, 2023). To detect these ideas, their method uses many models trained on Google Earth photos using unsupervised representation learning. They demonstrate the effectiveness of matching individual ideas to retrieve a scene or landscapes that resemble a satellite image of the Republic of Cyprus’s geographical area that the user has picked. Their findings highlight the advantages of segmenting the landscape similarity problem into discrete ideas strongly associated with remote sensing instead of using a single model that aims to target all underlying concepts.

Compared to earlier works in this section, the proposed approach contributes to developing an automated system for analyzing aerial images to extract meaningful land cover information using data-augmented and deep-learning approaches.

Dataset and preliminaries

We utilize a dataset that is both extensive and diverse, with manual annotation based only on three color channels (RGB) data, to allow for effective training of a model for precise semantic segmentation. For this purpose, this study used the LandCover.ai dataset (Boguszewski et al., 2021). The dataset images are categorized into four classes based on their relevance and importance for public administration purposes: building (1), woodland (2), water (3), and road (4). Buildings are considered objects that remain in a fixed location, excluding greenhouses. Each structure is only tagged for its roof and visible walls because our images are not full orthophotos. In contrast to single trees and orchards, woodland refers to an area where several trees grow in proximity. Water comprises both moving and still water, such as rivers and lakes, but not dry riverbeds or ditches. The road includes road and rail transport infrastructure, including parking lots and unpaved roads. Finally, background refers to areas not classified under any of the above classes, such as fields, grass, pavements, and other objects not included in the classes mentioned above. The different land used in the dataset is mentioned in Table 1. The purpose of Table 1 is to provide a clear and organized summary of the different land covers in the dataset, their corresponding area covered, and their assigned numerical label. The Class column lists the land cover type, such as Background, Building, Woodland, Water, and Road. Our study utilized label datasets with dimensions of 512 × 512 pixels. Each pixel in these label datasets represents one of five classes considered in our analysis. These classes were meticulously defined based on the specific criteria relevant to our study. The label datasets served as the ground truth for evaluating the performance of our proposed model, allowing for precise classification and segmentation tasks. The Coverage (km) column indicates the area covered by each type of land cover in kilometers. The Label column assigns a numerical label to each type of land cover.

Table 1 The different land covers the area.

Class	Coverage ( km2)	Label	
Background	125.75	0	
Building	1.85	1	
Woodland	72.02	2	
Water	13.15	3	
Road	3.5	4	

The dataset comprises photographs used to generate an aerial digital orthophoto of Poland. The reference data of the LPIS is continually updated using images collected from the public geodetic resource. Cartesian “1992” (EPSG:2180) coordinates were used to create the digital orthophotos. The images’ spatial resolution was either 25 or 50 cm per pixel, and they were taken using three spectral bands (RGB). These images were taken between 2015 and 2018, and the average flying season in Poland is between April and September. As a result, the obtained images cover various optical conditions at various times of the year and stages of the vegetative cycle. This data set is massive and versatile (Boguszewski et al., 2021).

To ensure maximum diversity, we carefully handpicked 41 orthophoto tiles from different counties across all regions of Poland, as illustrated in Fig. 1. Each tile covers an area of approximately 5 km2. Out of the 41 tiles, 33 have a resolution of 25 cm (roughly 9,000 × 9,500 pixels), while the remaining eight have a resolution of 50 cm (approximately 4,200 × 4,700 pixels), resulting in a total coverage of 176.76 and 39.51 km2 respectively, amounting to a total area of 216.27 km2. Figure 2 shows a selection of sample images.

Figure 1 Selection of orthophotos: Different locales exhibiting varied morphological, agrarian, and vegetative attributes, contributing to diverse dataset (Boguszewski et al., 2021).

Figure 2 Variability of selected images: Variations in regions, seasons, time of day, weather, and lighting conditions (Boguszewski et al., 2021).

Land cover characteristics

The chosen regions are situated in Central Europe, specifically in Poland. Poland is mainly located in the eastern part of the North European Plain, with a latitudinal extension of its geographic regions. As one moves from north and center towards the country’s south, the landscape gradually transitions from lowlands to highlands and mountains. The country’s terrain is predominantly agricultural, with varying agrarian structures covering about 60% of the area, and it also comprises coniferous, deciduous, and mixed forests that cover 29.6% of the land. Poland’s forest coverage is comparable to the average of Europe (excluding Russia) and North America, covering approximately 33% of the area. Coniferous woods are Poland’s most common forest type, accounting for 68.4% of the total forest area, with pine accounting for 58%. There are 38 urban areas with populations of 100,000 people or more, including one with over 1,000,000 people. In the north of Poland, there are vast postglacial lake districts, and numerous ponds are also present throughout the rest of the country.

Proposed approach

The proposed methodology of this work is presented in Fig. 3. The proposed methodology for the LandCover.ai dataset classification involves several key steps. Firstly, the dataset images and their corresponding masks were divided into 512 × 512 tiles, and smaller ones on the right and bottom edges were discarded. The tiles were then randomly shuffled and organized into the train, validation, and test sets. To further enhance the training set, offline augmentation was applied using imgaug, resulting in a total of 74,700 samples in the training set. The proposed methodology also includes using three different models for the segmentation task: the Vanilla UNet model, the UNet model with the Resnet50 encoder, and the DeepLabV3 model with the Resnet50 encoder. These models were defined using the PyTorch-based framework, SMP library, with varying parameters such as encoder name, encoder weights, decoder atrous rates, and encoder output stride. The proposed approach utilized accuracy, precision, recall, F1-score, model training and validation loss, and model predicted score to evaluate the performance of the deep learning models used in this work. The proposed methodology aims to improve the detection accuracy and robustness of the LandCover.ai dataset classification through appropriate data preprocessing and model selection.

Figure 3 Proposed approach for detecting land usage.

Data pre-processing

To begin with, we divided the 41 images and their corresponding masks into 512 × 512 tiles and discarded smaller ones on the right and bottom edges. The images were divided into 512 × 512 tiles, which likely improved memory efficiency and ensured compatibility with the LandCover.ai dataset, which may also use tiles of the same size. By discarding smaller tiles on the right and bottom edges, the authors ensured that all tiles used in the analysis were of uniform size, thus avoiding potential memory overflow issues during processing. Dividing the images into smaller tiles helps preserve finer details and features in the original images, which is crucial for accurate land cover classification.

Next, we randomly shuffled the tiles and organized them in the following manner: 15% (1,602) of the tiles were allocated to the test set, another 15% (1,602) were designated as the validation set, and the remaining 70% (7,470) were assigned to the train set. The tiles were randomly shuffled to prevent any biases in the data ordering. This organization ensures that the model is trained on a diverse data set and evaluated on unseen data, thus improving its generalizability. We have included lists of filenames along with the dataset.

Data augmentation

Appropriate augmentation that mimics various flying and land cover circumstances would be advantageous. As a result, we used an offline augmentation strategy with imgaug (https://imgaug.readthedocs.io/en/latest/) on the training set. For each tile, we added nine augmented copies with random variations in properties, including hue, saturation, grayscale, contrast, brightness, sharpness, noise addition, flipping, rotation, cropping, and padding. This method yielded a total of 74,700 samples in the training set.

Classification models

This section demonstrates the models for experimentation. We enlist all models and their parameter settings to make the study reproducible.

Vanilla UNet model: The UNet model used in the algorithm is defined in the utils module. The model = UNet(in_channels = 3, out_channels = 5).to(device) line of code creates an instance of the UNet model. The in_channels parameter specifies the number of input channels to the model, which is set to 3. The out_channels parameter specifies the number of output channels from the model, which is set to 5. The UNet model is designed to take in images with RGB and output a segmented image with five different classes.

UNet model with Resnet50 encoder: The second model used in the algorithm is the UNet model with a Resnet50 encoder. This model is defined using the SMP library, a PyTorch-based framework for training semantic segmentation models. The model = smp.Unet (encoder_name = “resnet50”, encoder_weights = “imagenet”, classes = 5).to(device) line of code creates an instance of the UNet model with the Resnet50 encoder. The encoder_name parameter specifies the type of encoder to use, which is Resnet50. The encoder_weights parameter specifies the pre-trained weights for the encoder, which are the weights trained on the ImageNet dataset. The classes parameter specifies the number of output classes for the segmentation task, which is set to 5.

DeepLabV3+ model with Resnet50 Encoder: The third model used in the algorithm is the DeepLabV3+ model with a Resnet50 encoder. This model is also defined using the SMP library. The model = smp.DeepLabV3Plus (encoder_name = “resnet50”, encoder_weights = “imagenet”, decoder_atrous_rates = (12,18,24), encoder_output_stride = 16, classes = 5).to(device) line of code creates an instance of the DeepLabV3+ model. The encoder_name and encoder_weights parameters are identical to the UNet model with the Resnet50 encoder. The decoder_atrous_rates parameter specifies the atrous rates for the decoder module of the model, which are set to (12, 18, 24). The encoder_output_stride parameter specifies the output stride of the encoder module, which is set to 16. The classes parameter specifies the number of output classes for the segmentation task, which is set to 5.

The Semantic Segmentation with Multiple Models algorithms aim to predict five land uses from input images using multiple models. The algorithm begins by loading and splitting the images into smaller images of size 512 × 512 pixels. Next, the data augmentation is applied to the images using the data_augmentation() function to enhance the images and use a combination of modifications to improve our training set. We can use data augmentation to recreate photographs taken in various flying and land cover situations and at various times of the year. The loss function and method are defined as the intersection over union (IoU) loss and the mean IoU loss, respectively. The SegDataset is created using f (mode, transforms, ratio), where mode is the mode of the dataset (train, test or val), transforms is the set of data augmentations, and the ratio is the proportion of the dataset to be used for training.

The Model function is used to train the model and returns the trained model. The PredictRatings function predicts ratings based on the trained model and returns the predicted ratings. The original data is loaded and split into input (X) and output (y) variables in the Main function. The Model function is called with the input and output variables as arguments, which trains and returns the trained model. The PredictRatings function is then called with the trained model and input variables to predict the ratings. The model’s accuracy is calculated using the accuracy equation, and the class probability of the models is calculated using the softmax function. Finally, the predicted ratings, accuracy, and class probabilities are returned.

Experimental setup

The experiments were conducted using Google Colaboratory, a cloud-based platform for developing and running Python code using Jupyter Notebook. The notebooks were executed on a Google Colab instance with a GPU runtime. The experiments used the following configuration: GPU: Nvidia Tesla T4, RAM: 12 GB, CPU: Intel Xeon 2.30 GHz, Python version: 3.7.12, and PyTorch version: 1.9.0.

Setting the training scheme

This section establishes the parameters of the training approach taken throughout this Notebook. Local experiments were used to determine the optimal values for the hyperparameters. Since we could not run many experiments due to GPU restrictions, the hyperparameters we settled on should not be considered optimum. The IoU measure, defined as follows, is used to check the accuracy of our models.

It is possible to express an image mask y as a 2D array of dimensions H × W. The prediction mask for semantic segmentation can also be represented as a 2D array of size H × W. Let us pretend that C is the total number of categories in the range 0,1,…C−1. Fix an index 0≤k≤C−1. Then, the number of pixels on mask y, which is equal to k, can be described by the cardinal number of the set (i,j):1≤i≤H,1≤j≤W,y[i,j]=k. Below Eq. (1) shows the formula of cardinal numbers.

(1) Yk=∣(i,j):1≤i≤H,1≤j≤W,y[i,j]=k∣

Similarly, for the prediction mask y′, the number of pixels equal to k is given by following Eq. (2).

(2) Y′​k:=|(i,j):1≤i≤H,1≤j≤W,y′[i,j]=k|

The intersection over union for the class k on the image with mask y is calculated as following Eq. (3):

(3) Mk=1N∑i=1N|Yk,i∩​Y′k,i||Yk,i∪​Y′k,i|

where |A| denotes the cardinality of set A. The value of Mk is always between 0 and 1. Values close to 0 indicate few correct predictions, while values close to one indicate that almost all predictions are correct. For a finite sequence of images, masks, and predictions, the mean IoU (MIoU) is given by following Eqs. (4) and (5):

(4) IoU=1C∑k=0C−1Mk

(5) MIoU=1C∑k=0C−1Mk

Now we can compute the IoU loss (also known as the Jaccard loss) by following Eq. (6):

(6) IoULoss=1−IoU=1−1C∑k=0C−1Mk.

We employ the IoU loss, the standard metric for validation on semantic segmentation tasks, to train our models. Here, we employ Segmentation Models Pytorch’s Jaccard loss implementation.

Algorithm 1 Semantic segmentation with multiple models.

Require: landcoveraidataimages	
Ensure: Predicting5Landuses	
 1: Evaluation Metrics: Accuracy, precision, recall, F1-score	
 2: Load images from the data root directory and visualize some samples	
 3: Split_images(desired_size = 512) = 512 × 512	
 4: Initialize the Data augmentation using the function DA = data_augmentation()	
 5: Define loss function;	
 6: Define Loss method; IoULoss=1−IoU=1−1C∑k=0C−1Mk	
 7: SegDataset=f(mode,transforms,ratio) where:	
mode; is the mode of the dataset (train, test, val)

transforms=DA; is a set of data augmentations applied to the dataset

ratio is the proportion of the dataset to be used for training

	
 8: function Model (X,y)	
 9:   Start Model training	
10:   Model(data, IoU Loss, loss_fn, mod_epochs =1, regularization = “L2”, reg_lambda = 1e-6, early_stopping = True, patience = 4, verbose = True, save = True, stopping_criterion = ”loss)	
11:   return model	
12: function PREDICTRATINGS(model, X)	
13:    Start Model Prediction	
14:    return ypredicted	
15: procedure MAIN	
16:   data← Original data	
17:   X,y← Split_data(data)	
18:   model← Model( X,y)	
19:   ypredicted← PredictRatings( model,X)	
20:   Calculate the accuracy of the model (Accuracy Equation: accuracy=TP+TNTP+TN+FP+FN)	
21:   Calculate the class probability of the models by the equation; class_probs(x)i,j,k=exp⁡(model(x)i,j,k)∑_c=1Cexp⁡(model(x)_c,j,k)	
22:   return ypredicted,accuracy,class_probs	

Experimental analysis and results

The experimental results of the proposed approach are explained in this section. The study used the LandCover.ai dataset for the experiments. Three sections of the dataset were separated. The training, validation, and test sets were divided at a percentage of 70%, 15%, and 15% respectively. The training set’s pixel distribution for each class for 512 × 512 pictures. The experiments were performed using Google Colab due to its open-access GPU computation. The study used three deep learning models (Vanilla-UNet, ResNet50 UNet and DeepLabV3 ResNet50).

Table 2 shows the experimental results of three models on a given dataset: Vanilla-UNet, ResNet50 UNet, and DeepLabV3 ResNet50. These models are evaluated using performance metrics such as accuracy, precision, recall, and F1-score. The three models completed the training using the 28, 20 and 30 epochs.

Table 2 Experimental results of proposed approach.

Model	Accuracy	Precision	Recall	F1score	
Vanilla-UNet	91.31	93.10	92.10	92.50	
ResNet50 UNet	94.37	94.80	94.20	93.70	
DeepLabV3 ResNet50	94.77	95.20	95.60	95.40	

The Vanilla-UNet model achieves an accuracy of 91.31% with a recall of 92.10%, a precision of 93.10%, and an F1-score of 92.50%. Training and validation loss for the Vanilla-UNet model is depicted in Fig. 4. Training lasts for 28 iterations. The initial values for the training and validation losses are 0.6967 and 0.6628, respectively. The training and validation losses reduce as the model is trained for more iterations. Each epoch includes a report detailing the training and validation loss, with the best model having the lowest loss. An anti-overfitting method, EarlyStopping counts the number of epochs during which the validation loss has not decreased before stopping training. The training above uses an early ending criterion of 4, indicating that the training will be terminated if the validation loss does not decrease for four epochs in a row. At epoch 28, the early stopping criterion is met, and the training is stopped. The final training loss is 0.3141, and the final validation loss is 0.2961.

Figure 4 Training and validation loss curve visualization.

ResNet50 UNet outperforms Vanilla-UNet with an accuracy of 94.37% and a precision of 94.80%. However, it has a lower recall of 94.20% and an F1-score of 93.70%. Figure 4B shows the training and validation loss of the ResNet50 UNet model. In the first epoch, the training loss was 0.5436, and the validation loss was 0.3822. The validation loss decreased compared to the initial value, and the model was saved. In subsequent epochs, the training loss decreased, indicating that the model improved on the training data. The validation loss also decreased, indicating that the model generalized well to the validation data. The model was saved every time the validation loss decreased. In epoch 8, the validation loss increased compared to the previous epoch, which triggered early stopping (a technique to stop the training process early if the model is not improving anymore). The early stopping counter was set to 5, which means that if the validation loss did not decrease for five consecutive epochs, the training would stop. In epochs 12 and 15, the validation loss decreased compared to the previous epochs, and the model was saved again. However, after epoch 15, the validation loss stopped decreasing, triggering early stopping after five consecutive epochs. The final validation loss was 0.2241, which was achieved in epoch 20.

DeepLabV3 ResNet50 has the highest accuracy of 94.77% with the highest precision, recall, and F1-score of 95.20%, 95.60%, and 95.40%, respectively. Figure 4C shows the training and validation loss of the DeepLabV3 ResNet50 model. The validation loss decreases from 0.365216 to 0.218037 over 24 epochs, indicating that the model’s performance improved significantly during this period. The training loss also decreased from 0.5107 to 0.1951 during the same period. The training and validation losses then start to fluctuate around 0.2–0.3, and the early stopping criterion is met after 29 epochs because the validation loss does not improve further. However, the training and validation losses provide insight into the model’s performance during the training process and can be used to optimize the model’s hyperparameters and prevent overfitting.

A broader comparison of these models indicates that DeepLabV3 ResNet50 performs better than the other models in accuracy, precision, recall, and F1-score. However, ResNet50 UNet also performs well with relatively high precision, while Vanilla-UNet has the lowest performance among the three models. These results suggest that using a more sophisticated architecture, such as ResNet50 and DeepLabV3, can improve the segmentation performance, which can be useful for various applications such as object detection and medical imaging.

Comparative analysis

Table 3 presents the experimental results of the proposed approach and existing approaches for semantic segmentation of satellite images (Saha, 2022).

Table 3 Comparative analysis of proposed with existing approach results based on classes.

Classes	Probability prediction score	
Proposed approach results	Existing approach results	
Vanila-UNet	ResNet50 UNet	DeepLabV3 ResNet50	UNet	
Background	99.8	99.9	99.6	–	
Building	99.1	99.0	98.3	24.7	
Woodland	99.8	99.8	99.5	86.1	
Water	99.6	99.8	99.6	80.2	
Road	98.9	99.1	97.7	61	

The proposed models perform better than the existing approach. The results are shown for different classes, i.e., Background, Building, Woodland, Water, and Road. The probability prediction score of each model is also given in the table. For the Background class, all three proposed models (Vanilla-UNet, ResNet50 UNet, and DeepLabV3 ResNet50) have performed very well with probability prediction scores of 99.8, 99.9, and 99.6, respectively. However, the existing UNet model has yet to be evaluated for this class. For the Building class, the proposed models have achieved probability prediction scores of 99.1, 99.0, and 98.3 for Vanilla-UNet, ResNet50 UNet, and DeepLabV3 ResNet50, respectively. Vanilla-UNet has the highest probability prediction score for this class among the proposed models. On the other hand, the existing UNet model has achieved a probability prediction score of 24.7, significantly lower than the proposed models. For the Woodland class, all proposed models have achieved high probability prediction scores of 99.8, 99.8, and 99.5 for Vanilla-UNet, ResNet50 UNet, and DeepLabV3 ResNet50, respectively. The existing UNet model has also achieved a high probability prediction score of 86.1 for this class. For the Water class, the proposed models have achieved high probability prediction scores of 99.6, 99.8, and 99.6 for Vanilla-UNet, ResNet50 UNet, and DeepLabV3 ResNet50, respectively. The existing UNet model has also performed well, with a probability prediction score of 80.2 for this class.

For the Road class, the proposed models have achieved probability prediction scores of 98.9, 99.1, and 97.7 for Vanilla-UNet, ResNet50 UNet, and DeepLabV3 ResNet50, respectively. Among the proposed models, ResNet50 UNet has this class’s highest probability prediction score. On the other hand, the existing UNet model has achieved a probability prediction score of 61, which is lower than the proposed models. The proposed approaches have achieved higher probability prediction scores than the existing UNet model for all classes. The lowest probability prediction score among the proposed models is 97.7 for the DeepLabV3 ResNet50 model for the Road class.

Table 4 presents a comparative analysis of different approaches based on their classifiers, datasets, and performance metrics. It provides valuable insights into the state-of-the-art techniques for image classification or segmentation tasks. The approach proposed by Boguszewski et al. (2021) utilized the DeepLabv3+ classifier with output stride (OS) 4 and data augmentation techniques. The approach was evaluated on aerial image datasets and achieved a performance of 85.56%. The approach presented by Lee & Lee (2022) where employed the UNet classifier. The study focused on aerial image datasets and reported a performance of 77.8%. The work by Lee & Lee (2022) focused on aerial and satellite image datasets. The UNet classifier was used, and the reported performance was 91.4%. The approach proposed by Chen et al. (2022) utilized the multi-level aggregation network (MANet) classifier. The study focused on aerial image datasets; the reported performance was 87.09%. The proposed model in the current study. The classifier used was DeepLabV3 ResNet50, and the evaluation was conducted on aerial image datasets. The reported performance was 94.77%. Our approach outperforms model DeepLabV3 ResNet50 on aerial image datasets based on the performance.

Table 4 Comparative analysis of proposed with existing approach results based on datasets.

Authors	Classifiers	Datasets	Performance	
Boguszewski et al. (2021)	DeepLabv3+ OS 4 + augmentation	Aerial image datasets	85.56%	
Lee & Lee (2022)	UNet	Aerial image datasets	77.8%	
Lee & Lee (2022)	UNet	Satellite image datasets	91.4%	
Chen et al. (2022)	Multi-level aggregation network (MANet)	Aerial image datasets	87.09%	
Proposed model	DeepLabV3 ResNet50	Aerial image datasets	94.77%	

Conclusion

With the increase in the availability of satellite imagery, enormous amounts of data are being collected daily. This spatial imagery provides valuable information that various entities can use. However, manually analyzing each image to obtain insights into land cover takes time and effort. To address this issue, this study proposes using deep learning techniques to accurately and efficiently classify satellite imagery pixel by pixel for land cover analysis. This study used the LandCover.ai dataset for model experiments. The three most important deep learning models (Vanilla-UNet, ResNet50 UNet and DeepLabV3 ResNet50) are used in this study. The proposed models show better performance compared to the existing approach. The Vanilla-UNet model has an accuracy of 91.31%, ResNet50 UNet obtained 94.37% accuracy, and DeepLabV3 ResNet50 has 94.77% accuracy. DeepLabV3 ResNet50 performs better than the other two models regarding accuracy, precision, recall, and F1-score. In the future, we intend to collect more data on land cover and apply semantic segmentation.

Supplemental Information

Supplemental Information 1 Code for Land cover classificaiton.

Additional Information and Declarations

Competing Interests

Author Contributions

Data Availability

The authors declare that they have no competing interests.

Sidra Abbas conceived and designed the experiments, performed the experiments, analyzed the data, performed the computation work, prepared figures and/or tables, authored or reviewed drafts of the article, and approved the final draft.

Ahmad Almadhor conceived and designed the experiments, performed the experiments, analyzed the data, performed the computation work, prepared figures and/or tables, authored or reviewed drafts of the article, and approved the final draft.

Gabriel Avelino Sampedro conceived and designed the experiments, performed the experiments, analyzed the data, performed the computation work, prepared figures and/or tables, authored or reviewed drafts of the article, and approved the final draft.

Shtwai Alsubai conceived and designed the experiments, performed the experiments, analyzed the data, performed the computation work, prepared figures and/or tables, authored or reviewed drafts of the article, and approved the final draft.

Abdullah Al Hejaili conceived and designed the experiments, performed the experiments, analyzed the data, performed the computation work, prepared figures and/or tables, authored or reviewed drafts of the article, and approved the final draft.

Ľubomíra Strážovská conceived and designed the experiments, analyzed the data, performed the computation work, prepared figures and/or tables, authored or reviewed drafts of the article, and approved the final draft.

Monji Mohamed Zaidi conceived and designed the experiments, performed the computation work, prepared figures and/or tables, authored or reviewed drafts of the article, and approved the final draft.

The following information was supplied regarding data availability:

The data is available at Kaggle (https://www.kaggle.com/datasets/adrianboguszewski/landcoverai) and LandCover.ai.

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
