# Peer review of "Efficient geospatial mapping of buildings, woodlands, water and roads from aerial imagery using deep learning"

_PeerJ Computer Science, doi:10.7717/peerj-cs.2039_

## Round 0.1 · original submission · Major Revisions

Please make sure that reviewer comments are addressed. I understand that sometimes it is challenging to address all reviewer comments especially when one recommends a minor revision and the other recommends a major revision. Please treat this as a major revision so that Reviewer 2's comments on experimental design and validity of findings are sufficiently addressed and Reviewer 1's comments are also addressed.

**Language Note:** The review process has identified that the English language must be improved. PeerJ can provide language editing services - please contact us at [email protected] for pricing (be sure to provide your manuscript number and title). Alternatively, you should make your own arrangements to improve the language quality and provide details in your response letter. – PeerJ Staff

Reviewer 1 ·

Basic reporting

no comment

Experimental design

No comment.

Validity of the findings

no comment

Additional comments

The study used three different architectures of deep learning (i.e., Vanilla UNet, ResNet50 UNet and DeepLabV3 ResNet50) to manage to accurately and efficiently classify aerial imagery considering pixel by pixel approach for land cover analysis. Following the results of the research, the proposed models showed better performance compared to the existing approach. Besides, the results of the research indicated that the DeepLabV3 ResNet50 outperformed two other DL architectures regarding accuracy, precision, recall, and F1-score metrics. I recommend publishing this paper after addressing a few minor comments as follows:
1. Even though satellite images and aerial photographs are both used for observing and capturing images of the Earth’s surface, they differ in various aspects such as the method of capture, resolution, coverage, cost, and applications. Accordingly, using them interchangeably is not recommended. As a suggestion, use the same word inside of the manuscript text. For example, in the title, it is mentioned aerial imagery and in abstract it is mentioned satellite imagery. And the same thing between the photos and images.
2. It is not clear that the objective of the research is the classification or change detection. The introduction all talks about the change detection, but the objective of the paper based on the results and other sections is classification. Please modify the introduction section accordingly, or if the aim of the research is change detection, then all sections must be corrected and written from scratch and then needs major revision. Please be specific.
3. What are the advantages of dividing images/photos into smaller images? The authors mentioned that the 41 images/photos were divided into smaller images/photos with the size 512 x 512. The division was done because of the size of LandCover.ai dataset, they are also in 512 x 512 or there are other reasons?
4. Following the previous comment, the label dataset (i.e., ground truth datasets) was not explained. What is their size? (512 x 512), and if each pixel in the label dataset represents one type out of 5 classes the authors considered? If it is feasible, briefly explaining of the label datasets is recommended adding to the body of the manuscript.
5. There is also a need to modify the reference type inside of the text. The references must be separated from texts. As an example, in the first paragraph of Introduction section: … is a significant resource (Zhou et al. 2011; Pauleit and Duhme 2000; Ahmed et al. 2017; Gerard et al. 2010).

Annotated reviews are not available for download in order to protect the identity of reviewers who chose to remain anonymous.

·

Basic reporting

The manuscript presents deep learning methods for precise and rapid pixel-by-pixel classification of satellite imagery for land cover analysis, which would be a significant step forward in resolving this issue., however the paper seems an incremental work since its scientific novelty and contribution are very limited or the authors failed to explain its novelty and contribution. Therefore it can not be considered for its possible publication in its current form in this journal. However, the manuscript can be considered after a thorough revision by addressing the following concerns:

1. The scientific novelty and contribution should be explained, why it is novel?
2. The literature review section should be revised and extended.
3. It should also be clearly stated how the present work intends to improve the proposals of other authors. On the other hand, it is necessary to clarify some aspects of the experimental methodology in more detail to assess the real usefulness of the method better.
4. It should be compared to the existing datasets and should be validated. How one can trust it and use it in their future research without validation?
5. The quality of Figure seems to be satisfactory, further improve all captions
6. The methodology part needs a re-organization so that readers of the paper can understand.
7. The Results & Discussion section should be thoroughly revised comparing the study results to the state-of-the-art (SOTA) models. The created dataset needs to be compared to the existing SOTA datasets, too.

Experimental design

Experimental Design needs revision as per above comments

Validity of the findings

Validity of the results need to be improved as per given comments

Additional comments

NA

---

## Round 0.2 · accepted · Accept

The original Academic Editor is not available so I have taken over in my capacity as Section Editor.

Thanks to the authors for their efforts to improve the work. The current version successfully satisfied the reviewers. It could be accepted now.

Reviewer 1 ·

Basic reporting

no comment

Experimental design

no comment

Validity of the findings

no comment

Additional comments

I recommend publishing the current version of the manuscript.

·

Basic reporting

Considerable changes are made in the revised version. It looks much improved.

Experimental design

I am satisfied now with the experimental design of the revised article.

Validity of the findings

Validation on findings are acceptable.